
**Joint optimal operation of the South-to-North Water Diversion Project considering the evenness**
**of water deficit**
Bing-Yi Zhou[1], Guo-Hua Fang[1], Xin Li[1], Jian Zhou[1], Hua-Yu Zhong[1]
[1]College of Water Conservancy and Hydropower Engineering, Hohai University, Nanjing 210098,
China
*Correspondence to:* Guohua Fang. (hhufgh@163.com)
**Abstract.** Inter-basin water transfer project is the main measure to address the water deficit crisis
caused by uneven distribution of water resources. The current water transfer operation mainly tends to
be in areas with small water transfer costs and is prone to encounter the problem spatial and temporal
imbalances in water allocation. To address the aforementioned issues, this paper defines a Water
Deficit Evenness Index (WDEI) aimed at minimizing the regional differences in water scarcity and
sharing the pressure of water scarcity as a social demand objective. This index is incorporated into a
joint optimization model for the South-to-North Water Diversion project (J-SNWDP) in Jiangsu, which
comprises both the ecological objective of the total water deficit (TWD) and the economic objective of
the pumping water (PW). Further, the NSGA-III algorithm and multi-attribute decision-making were
applied to solve the model and obtain an optimal operation strategy. The results showed that: 1) The
WDEI defined in this paper can mitigate the synchronized water scarcity in certain water users. In
typical normal years (wet year and dry year), the Water Deficit Evenness Index shows a reduction of
94.2% (81.8%, 76.7%) compared to the historical operation strategy; 2) The optimized operation
strategy can significantly reduce TWD and PW by 82.06% (37.69%, 52.36%) and 45.13% (3.25%,
21.51%) compared with the historical values, respectively, which can improve the water supply
satisfaction and reduce the project cost. At the same time, the lake storage capacity of the optimal
operation strategy performs well, and the water transfer efficiency of the river is significantly improved.
3) In this paper, targeted optimal operation strategies and potential ways to secure the project tasks are
proposed for different natural flow. Overall, it is of great significance to study the water supply equity
in the Jiangsu section of the South-to-North Water Diversion Project to alleviate the concentrated water
deficit in Jiangsu Province and other similar regions.





**Key Words.** South to North Water Diversion Project, China; Water deficit Evenness index; multi-
objective optimization; NSGA-III; Multi-attribute decision-making.
**1 Introduction**
Water demand has been increasing rapidly in recent years with economic development and
population growth (Dolan et al., 2021; Liu and Yang, 2012). As the demand for water increases, the
availability of water resources for human use continues to decline, resulting in water scarcity, increased
risk of flood and drought disasters, and exacerbation of the conflict between water supply and demand.
These social issues have become one of the key factors constraining sustainable development and
environmental protection worldwide. (Denaro et al., 2017; Florke et al., 2018; Jiang, 2009; Li et al.,
2020; Ma et al., 2020; Wang et al., 2017; Zhao et al., 2016). Inter-basin Water transfer projects have
been widely constructed worldwide as an effective way to address water scarcity issues caused by
uneven distribution of water resources and improve their utilization efficiency (Sun et al., 2021). The
California State Water Project, the Colorado River Aqueduct (Lopez, 2018), the Senqu-Vaal transfer in
South Africa and Lesotho (Gupta and van der Zaag, 2008), the Snowy Mountains Scheme in
southeastern Australia (Pigram, 2000), the IBWD project of the Agrestic region of Pernambuco (Cirilo
et al., 2021; Neto et al., 2014), and other inter-basin water transfer projects have all effectively
alleviated water scarcity issues in various regions. China is home to approximately 18% of the global
population. However, the country's water resources account for only around 6% of the world's total.
This imbalance between population and water resources presents China with significant water resource
challenges. As a result, inter-basin water transfer projects have been more extensively constructed in
China, such as the well-known South-to North Water Diversion Project (Guo and Li, 2012), Yunnan
Central Water Diversion Project (Xiang et al., 2022), and so on. At least 10 % of the cities worldwide
receive water from IBWD projects (McDonald et al., 2014). Specifically, with an estimated investment
of around 78 billion USD, the South-to North Water Diversion Project (SNWDP) is regarded as the
largest inter-basin water transfer project in the world. The project runs along numerous water users, and
the water resources it provides have already benefited hundreds of millions of people, with even more
expected to be served in the future (Pohlner, 2016).
With the ongoing emergence of issues such as environmental pollution and degradation, global





climate change, and population growth, the problem of water scarcity has become increasingly

prevalent worldwide. Hence, effectively operating inter-basin water transfer projects and enhancing the

dispatching benefits is a challenging task. Currently, most IBWD projects primarily follow various

laws, regulations, policy guidelines, and historical experience in dispatching strategies set by the

government. However, there is a lack of detailed operating rules for different natural scenarios. Leading

to an imbalance in water supply across regions and putting some water users at high risk of water

scarcity. Addressing the aforementioned issues, there are considerable studies on the water resources

operating strategy of the supply-oriented IBWD projects in terms of social, economic, ecological, and

environmental (Gan et al., 2011; Liu and Zheng, 2002; Xu et al., 2013; Zhu et al., 2014). In general,

meeting the water demand of various users is the main task of the IBWD project, with the

consideration of minimizing water deficit in previous studies (Guo et al., 2020; Wang et al., 2008).

Rather than the total amount of water deficit, the crux of the problem may actually be the concentration

of water deficit in a certain period of time or region, which has not yet received sufficient attention and

remains a major challenge. Therefore, both the total and spatial-temporal distribution of water deficit

should be considered in the optimization process. (Xu et al., 2013). In addition, users' demands and

decision makers' benefits should be considered as priorities (Zhang et al., 2012), so minimizing

pumped water (PW) is a direct way to reduce costs. At the same time, the proportion of the amount of

abandoned water and the water withdrawn from the river in the process of water diversion should also

be considered as secondary considerations. In order to solve the above problems and define reasonable

objectives, existing studies are mainly carried out from water supply index and cost index. Liu and

Zheng define the ratio of regional water consumption to water availability as the water pressure to

reflect water supply reliability (Liu and Zheng, 2002). Guo et al., consider social benefits, maximum

power generation and environmental flow satisfaction, into account (Guo et al., 2018). In addition,

Ouyang and Iop set the minimum water power loss as a target to support reservoir operation in terms of

energy conservation (Ouyang and Iop, 2018). However, due to the data on natural water and user water

demand as the determining factors of the operation strategy, and the obvious regional differences, most

of the objectives determined by the existing studies can only solve small-scale projects, otherwise it

would lead to failure. Xi et al., used the rainfall forecast information from the Global Forecast System

(GFS) and calculated user water demand by ration, and found that the resulting operation strategy

couldn't be effectively compared with the historical operation strategy, because it is impractical to

apply these objectives to guide operation (Xi et al., 2010).

The China's South-to-North Water Diversion Project (SNWD), as the world's largest inter-basin

water transfer project, has provided 30.6 $10^8 m^3$ of water to the Hai River Basin since its official
operation in 2013. This has significantly alleviated the water supply deficit of large- and medium-sized
cities along the Beijing-Tianjin-Hebei-Henan route, accounting for 70 % and 90 % of the domestic
water in Beijing and Tianjin, respectively, demonstrating remarkable benefits (Liu et al., 2013). In
recent years, experts and scholars have extensively discussed the impact of the SNWDP on ecological
environment(Hu et al., 2022), changes in groundwater storage in the North China Plain(Zhang et al.,
2021), project benefits(Yang et al., 2021), and water quality (Wang et al., 2016), among other issues.
However, as the project continues to operate, it is necessary to shift the focus to dispatch management
in order to enhance the sustainability of the project. The SNWDP connects China's four major river
basins: the Yangtze River, Yellow River, Huai River, and Hai Ricer, involving multiple provinces such
as Shandong, Jiangsu, and Anhui, and presenting a highly complex and dynamic water situation,
especially in the Jiangsu section (Vogel et al., 2015). The project utilizes regulating reservoirs, sluice
stations, and pump stations to connect numerous water users along the route, supplying water to
various water-consuming sectors from both sides of the canal. However, due to differences in the
location and timing of natural inflows and water users, an imbalance in water supply has arisen. The
actual operation of the water supply plan is usually implemented under the guidance of the government
authorities, with reference to historical dispatch strategies. This approach may lack objectivity and
accuracy, potentially leading to inefficiencies in water resource utilization and the ineffectiveness of
operation strategies (Chen et al., 2019; Peng et al., 2015; Sheng et al., 2020). At present, there have
been some studies attempting to address this issue, but they tend to focus on meeting the total water
demand and improving the overall benefits (Li et al., 2017; Zhuan et al., 2016), neglecting the fairness
of water supply among different regions. As a result, water supply may become concentrated on a
specific user or time period. Therefore, it is of great theoretical significance and practical application
value to establish a scientific and systematic optimal operation model, to quantitatively analyze the
water resources allocation in users, to reflect the sophisticated water diversion process to guarantee
water supply, and to give full play to the comprehensive benefits of the IBWD project (Nazemi and
Wheater, 2015).

To address the above problem, this paper studies the Jiangsu section of South-to-North Water



Diversion project (J-SNWDP). The three main contributions of this paper are as follows: 1) defines the
Water Deficit Evenness Index (WDEI), and incorporates it into the optimization model together with
the Total Water Deficit (TWD) and the Pumped Water (PW), to meet the requirements of both decision-
makers and users; 2) incorporates the amount of abandoned water and the water withdrawn from the
Yangtze River into the decision indicator set, and uses the multi-attribute decision making method to
filter the Pareto front strategies of NSGA-III and finds the optimal operation strategy that balances the
economic and ecological benefits; 3) the paper compares the optimal operation strategy selected in
three typical years (wet year, normal year, and dry year) with the historical operation strategy under the
same natural conditions to verify the superiority of the optimization results, and puts forward
reasonable optimization suggestions for the SNWD project.
The paper is structured as follows: Section 2 presents the study area; Section 3 presents materials
and methods; Section 4 presents the results and discussion; Section 5 draws conclusions.

## 2 Materials and methods

### 2.1 Study area

#### 2.1.1 Regional Overview

The Jiangsu section of the South-to-North Water Diversion Project (J-SNWDP) crosses the
Yangtze River and Huai River basins. The simplified map of the J-SNWDP is shown in Fig. 1. The
Beijing-Hangzhou Grand Canal runs through the north and south of Jiangsu Province, connecting the
Yangtze River basin, the Huai River basin and the Yishusi River basin. The total length of the Jiangsu
section is 404 km, along which six cities are involved, namely Yangzhou, Huaian, Yancheng, Suqian,
Lianyungang and Xuzhou. The J-SNWDP consists of 3 impounded lakes (Hongze Lake, Luoma Lake,
Nansi Lake), 6 sluices (e.g., Erhe sluice, Gaoliangjian sluice, etc.), and 13 pumping stations (Huaian
Station, Jiangdu Station, etc.), forming a double-route water diversion system including the West Canal
and the East Canal Route. The water supply scope of the whole J-SNWDP covers three provinces of
Jiangsu, Anhui, and Shandong, supplementing the agricultural, industrial and domestic water supply as
well as navigation and ecological water supply in the areas along the water transfer route. The scale
parameters of the pumping stations and sluices in J-SNWDP are listed in Table 1. The characteristics of
the lakes in the J-SNWDP are shown in Table 2.

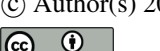



**Table 1 Pumping stations and sluices of the Jiangsu section of the South-to-North Water Diversion Project**

|  | Number | name | Scale (m³/s) | Number | name | Scale (m³/s) |
|---|---|---|---|---|---|---|
| Pumping station | P1 | Jiangdu | 100 | P8 | Sihong | 120 |
|  | P2 | Baoying | 400 | P9 | Suining | 110 |
|  | P3 | Jinhu | 400 | P10 | Zaohe | 175 |
|  | P4 | Huaian | 300 | P11 | Pizhou | 110 |
|  | P5 | Hongze | 150 | P12 | Liushan | 125 |
|  | P6 | Huaiyin | 300 | P13 | Taierzhuang | 125 |
|  | P7 | Siyang | 230 |  |  |  |
| Sluice | S1 | Nanyunxi | 400 | S4 | Yangzhuang | 500 |
|  | S2 | Gaoliangjian | 500 | S5 | Yanhe | 500 |
|  | S3 | Erhe | 500 | S6 | Huaiyin | 500 |

**Table 2 Lakes of the Jiangsu section of the South-to-North Water Diversion Project**

| Lake name |  | Hongze | Luoma | Nansi |
|---|---|---|---|---|
| Dead lake level (m) |  | 11.30 | 21.00 | 31.30 |
| Normal lake level (m) | Flood season | 12.50 | 22.50 | 32.30 |
|  | Non-Flood season | 13.50 | 23.00 | 32.80 |
| Regulation storage (10⁸ m³) | Flood season | 15.30 | 4.30 | 4.94 |
|  | Non-Flood season | 31.35 | 5.90 | 8.00 |
| Monthly range lake level for water diversion (m) | Jul - Aug | 12.00 | 22.22-22.10 | 31.80 |
|  | Sep - Oct | 12.00-11.90 | 22.10-22.20 | 31.50-31.90 |
|  | Nov - Mar | 12.00-12.50 | 22.10-23.00 | 31.90-32.80 |
|  | Apr - Jun | 12.50-12.00 | 23.00-22.50 | 32.30-31.80 |

When the J-SNWDP is supplying water along the water transfer route, users closer to the water source (such as JBHD User, S-S User, LM User, etc.) are generally prioritized. On the other hand, users like the Feihuanghe User, Lianyungang User, and Siyang-Zaohe User, which are located farther from the water source and receive less water from Luoma Lake, often require water to be pumped from the Yangtze River and Hongze Lake for replenishment. Combined with the uneven spatial and temporal distribution of precipitation, these recipient zones are more susceptible to water shortages.

In Fig. 2, the complex relationship of the J-SNWDP is generalized into a schematic diagram according to the geographical location. In order to represent the main components of the system and the connection between the backbone rivers, we regard lakes, pumping stations, sluices as the nodes and





the backbone river as the connecting line, while the water users are reasonably distributed among the
nodes of the water transmission route.
The joint optimal operation process of the J-SNWDP is described as follows. In addition to natural
precipitation and lake inflow, the Yangtze River is the main water source of water, which is pumped
from the river to the West Route and the Canal Route by P1 pumping station (see Fig. 2). The West
Route is pumped step by step from P2, P3, and P4 to Hongze Lake through Sanyang River, and Jinbao
Channel, and pumped from P9, P10, and P11 to the north through Xuhong and Fangting Rivers. The
Canal Route along the Beijing-Hangzhou Grand Canal route is pumped north from P5, P6, P7, and P8.
The two water transmission routes supplement the water demand of users along the routes while
pumping to the north, and merge into one line at the intersection of Fangting River and Zhongyun River.
It is then pumped further north until P12 and P13 are transported to Nansi Lake via Bulao and
Hanzhuangyun Rivers, respectively, thus delivering water outside the Jiangsu province. When the
natural inflow is high, the Hongze Lake can be discharged via P1 on the West Route, or through S2, S3,
S4, S5, and S6 to the relevant rivers and users. Luoma and Nansi Lakes are discharged through the
original water transmission routes.

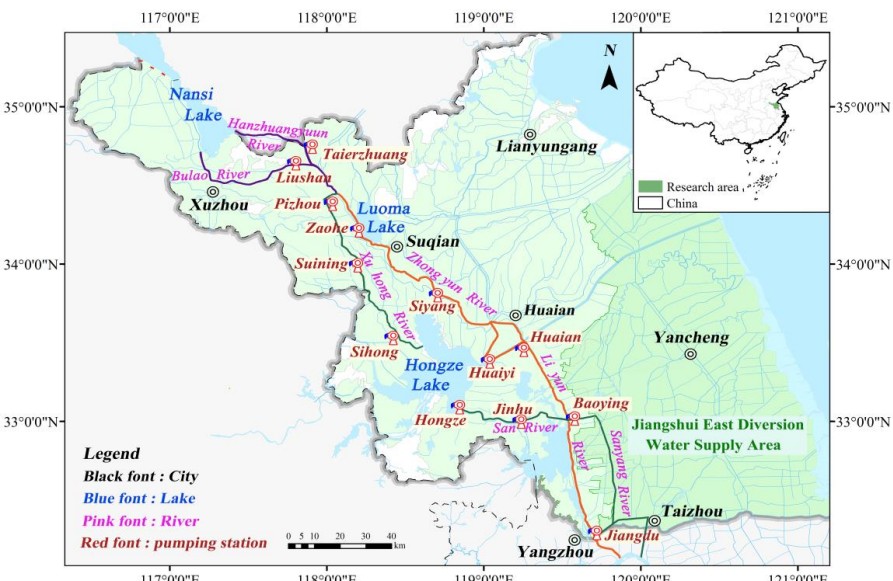


**Fig. 1. The Jiangsu section of the South-to-North Water Diversion Project. The orange, green and purple**
**lines represent the Canal Route, the West Route, and intersection of the two routes to transport water**
**outside the province, respectively.**





**Fig. 2. Schematic diagram of the Jiangsu section of the South-to-North Water Diversion Project.**





**2.1.2 Data**

The joint optimal operating model of the J-SNWDP uses a monthly time step. According to the annual natural inflow data of Hongze (HZ), Luoma (LM) and Nansi (NS) Lake since 2013 from Jiangsu Water Resources Bulletin, three hydrological years were selected to represent wet (2017.10-2018.09, annual mean inflow: $465.12 \times 10^8$ m³), normal (2019.10-2020.09, annual mean inflow: $172.80 \times 10^8$ m³), and dry (2013.10-2014.09, annual mean inflow: $103.06 \times 10^8$ m³) years, respectively. The typical annual runoff curves of HZ, LM and NS Lakes are shown in Fig. 3. The water supply area of the J-SNWDP is divided into 13 receiving areas according to geographical location, and water demand data is provided by the Jiangsu Provincial Water Resources Department: wet years (2017.10-2018.09, water demand: $171.29 \times 10^8$ m³), normal years (2019.10-2020.09, water demand: $173.51 \times 10^8$ m³), normal years (2013.10-2014.09, water demand: $181.75 \times 10^8$ m³).

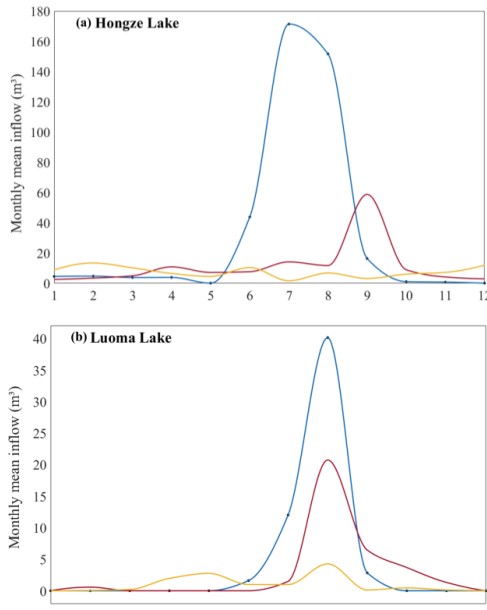

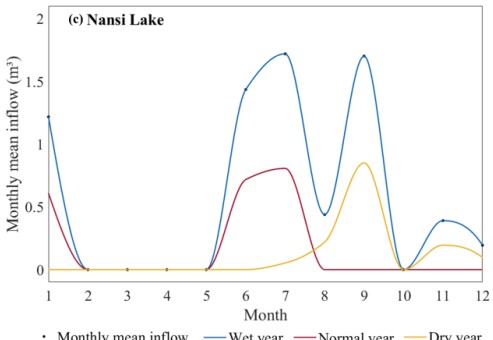


**Fig. 3. Monthly mean inflow curves of the (a) Hongze, (b) Luoma, (c) Nansi lakes in typical wet, normal, and**
**dry years.**
**2.2 Defining water deficit evenness index**

For IBWT projects, issues of equitable water supply often arise, especially for projects like the J-

SNWDP, which has 13 water users along its route. The issue is particularly severe here. Different users
have significant spatial variations in location, as well as large differences in water supply costs. Luoma
Lake, which almost receives no inflow during dry periods yet bears the primary responsibility for out-
of-province water supply task, requires multilevel pumping stations to pump water upward (water
needs to be pumped from Luoma Lake through two pumping stations, from Hongze Lake through six
pumping stations, and from the Yangtze River through nine pumping stations), creating a substantial
burden on water supply. In past water supply dispatching of the J-SNWDP, the pursuit of maximum
benefit while neglecting this issue would obviously cause certain users to bear severe water shortage
risks, resulting in significant damage to the water supply system. Therefore, considering the fairness of
water supply is very important for the practical operation of the project. Hence, the WDEI (Water
Deficit Evenness Index) is defined as an indicator representing the degree of water deficit
concentration, and the variance van reflect the difference in water shortages among various recipient
zones. By incorporating the WDEI index into the optimization objective and minimizing it, the
difference in water shortage can be reduced as much as possible. So the WDEI is below:
$$WDEI = \min \frac{\sum_{t=1}^{n} \left( QR(i,t) - \frac{\left( \sum_{i=1}^{J} QS(i,t) \right)}{J} \right)^2}{P} , \qquad (1)$$





**2.3 The joint optimal operation model of J-SNWDP**

Herein, the joint optimal operating model of the J-SNWDP was constructed with 228 decision variables (13 pumping stations and 6 sluices). The objective function and associated constraints are formulated as follows.

*2.3.1 Objectives*

(1)Minimizing the total water deficit (TWD)

TWD is a measure of how well the operation strategy completion is being implemented. This objective aims to minimize the total amount of water deficit at the end of a given operation period, potentially improving the satisfaction of water demand for users and increasing the operation strategy completion.

$$TWD = \min \sum_{t=1}^{T} \sum_{i=1}^{n} QR(i,t), \tag{2}$$

where $QR(i, t)$ is the water deficit of the $i$ th user at time step $t$, $i= 1, 2, …, n$, with $n$ being the total number of users, $t= 1, 2, …, T$, with $T$ being the whole operating period.

(2)Minimizing water deficit evenness index (WDEI)

WDEI indicates the degree of concentration of the water deficit and can be used as an indicator of the uniformity of water diversion. The lower the WDEI, the better the strategy is.

$$WDEI = \min \frac{\sum_{t=1}^{n} (QR(i,t) - \frac{\sum_{i=1}^{n} QR(i,t)}{n})^2}{J}, \tag{3}$$

where $P$ is the total number of pumping stations ($P= 13$ in this study).

(3)Minimizing pumped water (PW)

PW reflects the economy of operation strategy. The lower the PW is, the less the operating costs.

$$PW = \min \sum_{t=1}^{T} \sum_{p=1}^{P} QS(p,t), \tag{4}$$

where $QS(j, t)$ is the water pumped by the $p$th pumping station at time step $t$, $p= 1, 2, …, P$.

*2.3.2 Constraints*

Systems operation should obey operating rules and physical constraints, such as water balance, pumping capacity, and lake storage constraints. The mathematical expressions of the constraints are shown as below.



(1)Water balance constraint
The water balance constraint should be satisfied in the water diversion process.

$$V(i,t+1)=V(i,t)+Q(i,t)+DI(i,t)+PC(i+1,t)-DO(i,t)-W_1(i,t)-PR(i,t),\quad(5)$$

At time step $t$, where $Q(i, t)$ is the inflow of the $i$ th lake; $W_1(i, t)$ is the water demand of the $i$ th lake
(water to be supplemented by SNWD project after deducting the locally available water); $DO(i, t)$ is
water diversion to the north from the $i$ th lake; $DI(i, t)$ is the water pumped into the $i$ th lake; $PC(i, t)$ is
the water discharged into the $i$ th lake; $PR(i, t)$ is the water discharged from the $i$ th lake.
(2)Pumping capacity constraint

$$\begin{aligned}0\le DO(i,t)\le DO_{\max}(i,t)\\0\le DI(i,t)\le DI_{\max}(i,t)\end{aligned},\quad(6)$$

At time step $t$, where $DO_{max}(i, t)$ is the maximum pumping capacity that is pumped into the $i$ th lake;
$DI_{max}(i, t)$ is the maximum pumping capacity that is diverted north from the $i$ th lake.
(3)Sluice capacity constraint

$$0\le PR(i,t)\le PR_{\max}(i,t),\quad(7)$$

where $PR_{max}(i, t)$ is the maximum sluice capacity at time step $t$.
(4)Lake storage constraint

$$V_{\min}(i,t)\le V(i,t)\le V_{\max}(i,t),\quad(8)$$

where $V_{min}(i, t)$ and $V_{max}(i, t)$ are the minimum and maximum water storage capacities at time step $t$,
respectively. When $V(i,t)<V_{\min}(i,t)$ is water deficit, $QR(i,t)=V_{\min}(i,t)-V(i,t)$, ensure that
the lake level is above the limit level; when $V(i,t)>V_{\max}(i,t)$ is abandoned water, ensure that the
water storage of the lakes is within a reasonable range.
(5)Minimum lake levels for water diversion

$$Z_{\min}(i,t)\le Z(i,t)\le Z_{\max}(i,t),\quad(9)$$

where $Z_{min}(i, t)$ and $Z_{max}(i, t)$ are the minimum and maximum level of the $i$ th lake at time step $t$,
respectively.
**2.4 Model solving and solution selection**
Considering the complex objectives and various physical constraints, the NSGA-III algorithm





(Deb and Jain, 2014) is used in this paper to solve the model in this paper. NSGA-III has been widely
used to solve various water-resource optimal diversion problems and to obtain optimal operation
strategies with the advantages of fast execution speed and high efficiency (Ni et al., 2019; Tang et al.,
2021; Zhou et al., 2020). The NSGA-III is used to solve the joint optimal operation model of the J-
SNWDP under three typical conditions: wet year, normal year and dry year. The model takes the actual
value of the pumped water of each pumping station as the decision variable. The population size,
generation, crossover rate and mutation rate are set to 200, 20000, 0.9 and 0.1, respectively.
After obtaining the set of operation strategies using NSGA-Ⅲ, this paper further applies multi-
attribute decision-making methods to screen and determine the optimal operation strategy. The
abandoned water reflects the regulation and storage capacity of the lakes, and the water withdrawn
from the Yangtze River reflects the impact of the water transferred outside the system on the operation
strategy. A multi-attribute decision indicator set is constructed, which includes the above two indicators
and the three optimization objectives of the model. Indicators are divided into water use efficiency and
cost of water use indicators as shown in Fig. 4. In this paper, we adopt the Analytic Hierarchy Process
(AHP) method to determine the subjective weights and the entropy weighting method to determine the
objective weights, further the combination weights are obtained by linear weighted average.

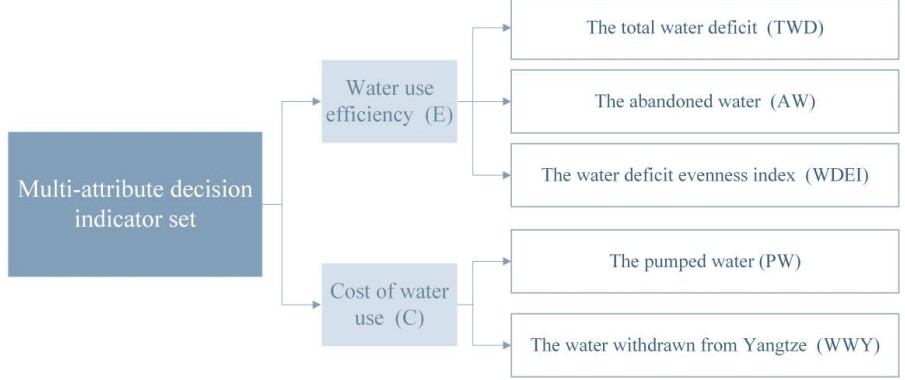

**Fig. 4. Determining the set of indicators for selecting optimal operation strategy in typical wet, normal, and**
**dry years.**
**3 Results**
**3.1 Pareto front strategies of NSGA-III under different years**
The historical operation strategy of the J-SNWDP follows the principle of 'replenishing the lake





during water shortage, and discharging during water surplus', with the primary goal of ensuring the
completion of water supply tasks. Firstly, the simulation of a single lake is carried out, and then the
simulation of each storage lake is carried out according to the top-down order, so as to complete the
conventional operation of the engineering system. Such a method is simply "robbing Peter to pay Paul".
The external water transfer is not fully deployed, but increases the operating costs. In contrast, the joint
optimal operation exhibits several advantages, such as controlling the minimum WDEI of each user,
ensuring that even if the total water shortage and the historical operating strategy are the same, the
water shortage pressure is evenly distributed to avoid undesirable water shortage concentration. In
addition, the optimal operation process pays more attention to the water storage function of the lake. In
the non-flood season, the water level of the reservoir is kept at the normal water storage level as much
as possible, and the water storage (discharge) is appropriate according to different natural inflow, so
that the whole system can realize the lake storage in the flood season and non-flood season at the same
time, and improve the utilization rate of water resources while fulfilling the water supply task.


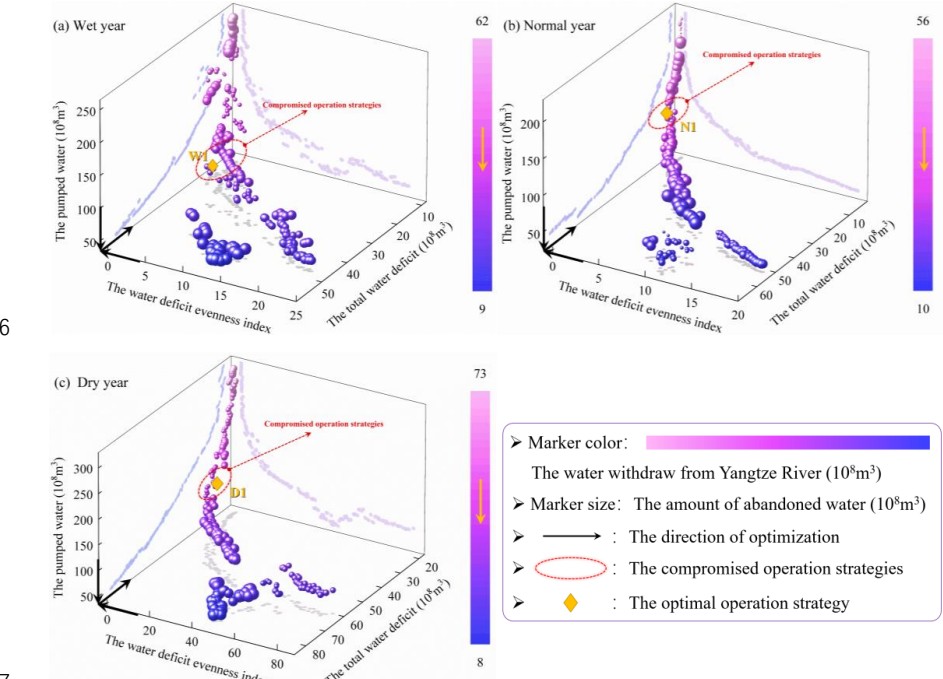

**Fig. 5. Illustration of Pareto front strategies for (a) wet, (b) normal, and (c) dry years. (W1, N1 and D1**
**represent the location of the optimal operation strategy in different typical years.)**
Fig. 5 provides visualizations of the optimal operation strategy based on Pareto sets in different





years. Obviously, the compromised operation strategies are well distributed in three typical years, and
the compromised operation strategies and the optimal operation strategy are located in the middle of
the curve. The relative relationship between the optimization objectives has been analyzed, the total
water deficit has a positive correlation with the water deficit evenness index. In contrast, the total water
deficit and water deficit evenness index have a negative correlation with the pumped water, indicating
that the water deficit gradually decreases with the increase of PW. The distribution pattern of the
marker is formed under the interaction of three objectives (see Fig. 5).
Fig. 5 (a) shows the competitive relationships between TWD, WDEI and PW in wet years that PW
decreases sharply and WDEI increases slowly at the beginning when TWD increases by $20\times10^8$m$^3$,
WDEI increases by about $3\times10^8$m$^3$, and PW decreases by about $164\times10^8$m$^3$, but when TWD exceeds
$31.33\times10^8$m$^3$, WDEI increases sharply and PW decreases slowly with TWD increasing by $20\times10^8$m$^3$,
WDEI increasing by about $9\times10^8$m$^3$, and PW decreasing by about $51\times10^8$m$^3$. The above relationships
obey the law of diminishing marginal utility. As a part of PW, the water withdrawn from the Yangtze
River represented by the marker color needs to be appropriately pumped according to the natural flow
and water demand of users. Generally, the proportion of water withdrawn from the Yangtze in PW
cannot be too high, affecting the lake storage capacity and pumping (operating) cost. Therefore, the
middle part or the lighter marker color tend to represent better results in the figure. Meanwhile, to
reduce the waste of water resources, the amount of abandoned water represented by the marker size
should be reduced as much as possible. Therefore, the yellow marker point is obtained as the optimal
operation strategy to compare with the historical operation strategy by multi-attribute decision-making.
The weights of decision indicators and the results of the comparison are shown in Table 3 and Table 4.
Similarly, the Pareto front strategies in Fig. 5 (b) and (c) also follow the above rules, but due to the
difference in natural inflow, the specific data are also quite different (see Table 4).
**Table 3 Determining the weights of indicators for selecting optimal operation strategy in typical wet, normal,**
**and dry years.**

| Evaluation indicators | Wet year | Normal year | Dry year |
|---|---|---|---|
| The total water deficit | 0.353 | 0.366 | 0.402 |
| The abandoned water | 0.194 | 0.203 | 0.264 |
| The water deficit evenness index | 0.207 | 0.248 | 0.215 |
| The pumped water | 0.134 | 0.101 | 0.074 |





| | The water withdrawn from Yangtze | | 0.111 | 0.080 | 0.045 |
|---|---|---|---|---|---|

**3.2 Comparison with historical operation strategy**

In order to verify the rationality of the strategies obtained by the joint optimal operation model, this section compares the optimal operation strategy selected in different typical years with the historical operation strategy.

**Table 4 Comparison of the main operation performance indicators of the historical and optimal operation strategy in typical years. (units: 108 m3)**

| Typical year | Scenario | The total water deficit | The water deficit evenness index | The pumped water | The amount of abandoned water | The water withdrawn from Yangtze |
|---|---|---|---|---|---|---|
| Wet | Historical | 65.77 | 22.87 | 119.27 | 374.56 | 25.74 |
| | Optimal | 31.33 | 4.16 | 93.62 | 310.91 | 23.65 |
| | **Decrement** | **52.36%** | **81.82%** | **21.51%** | **16.99%** | **8.12%** |
| Normal | Historical | 75.19 | 26.18 | 184.06 | 56.53 | 42.07 |
| | Optimal | 13.49 | 1.51 | 100.99 | 25.17 | 30.08 |
| | **Decrement** | **82.06%** | **94.24%** | **45.13%** | **55.47%** | **28.50%** |
| Dry | Historical | 63.22 | 21.54 | 159.88 | 13.60 | 26.62 |
| | Optimal | 39.39 | 5.01 | 154.68 | 0.05 | 37.34 |
| | **Decrement** | **37.69%** | **76.72%** | **3.25%** | **99.63%** | **-40.27%** |

Table 4 shows the optimal historical operation strategy in typical years. Compared with the optimal operation strategy, the TWD and WDEI of the historical operation strategy are very high, which means that the water deficit concentration in the historical operation strategy leads to the inability to reduce the TWD. Meanwhile, the PW and abandoned water are slightly large, indicating that the historical operation strategy has more disadvantages on economic and ecological benefits. WDEI can directly reflect the difference of water deficit among users. WDEI has sound optimization effects, of which the reduction in the optimal operation strategy was 94.2% (81.8 %, 76.7%) of the historical values in the typical normal year (wet year and dry year). The optimized operation strategy can significantly reduce TWD and PW by 82.06% (37.69%, 52.36%) and 45.13% (3.25%, 21.51%) compared with the historical values, respectively, while maintaining a very low WDEI. It has been demonstrated to be an excellent strategy for inter-basin water diversion, which can make multiple users share the risk of water deficit and alleviate the problem of water deficit concentration. The reduction of





the abandoned water and the water withdrawn from the Yangtze River represents an increase in the lake
storage capacity. The amount of abandoned water of the operation strategy is greatly reduced, and the
amount of abandoned water is only $0.051 \times 10^8 m^3$ in the dry year. This indicates that the optimal
operation can maximize the utilization efficiency of the limited water resources. However, the of
amount of water withdrawn from the Yangtze River increased by $10.72 \times 10^8 m^3$ after optimization in dry
years, indicating that the proportion of the water withdrawn from Yangtze River in PW was raised in
the optimal operation strategy to ensure uniform distribution of water and higher economic benefits.
The comparison of water deficit in water users between the optimal operation strategy and the
historical operation strategy in three typical years is illustrated in Table 5, which more intuitively
reflects the optimization effect of WDEI. The 'Variance' and 'Decrement' represent the difference
between the maximum and minimum values and the percentage reduction, respectively. The study
found that after optimization, the variance of the users was reduced by 62.93 % (78.07 %, 54.09 %) in
different typical years (i.e., bold font in Table 5). FHH user, S-Z user and HZH user, which have large
water deficit in the historical operation strategy, decrease by 14.63 (14.41, 14.69) $\times 10^8 m^3$, 5.61 (9.33,
1.77) $\times 10^8 m^3$ and 0.98 (7.98, 5.26) $\times 10^8 m^3$ of the optimal operation strategy, respectively (see Table 5).
The possible reason for the large water deficit of users is that they are far away from the main lakes and
rivers, or there are many users on the water transmission route that need to be supplied. Users jointly
share the water deficit risk in the optimal operation strategy in the case of little natural flow. The
difference between the maximum and minimum water deficit is controlled within $8 \times 10^8 m^3$, where the
difference in the normal year is only $3.13 \times 10^8 m^3$, and the problem of water deficit concentrated in a
particular user is alleviated. For example, in the dry year, the water deficit of users under optimal
operation strategy ranges from 0 to $7.64 \times 10^8 m^3$. Although the water deficit of GLH-HA user, LMH
user, and WSH user is slightly higher after optimization, the total water deficit is much lower than the
historical value and isn't concentrated in FHH user, HZH user, and S-Z user, compared with the results
of historical operation. Overall, the optimal operation strategy is more reasonable, and may accord with
the aspirations of both the government and the general public.
**Table 5 Comparison of water deficit in water users for (a) wet, (b) normal, and (c) dry years.**

| Typical years | Wet year | | Normal year | | Dry year | |
|---|---|---|---|---|---|---|
| Users | Historical | Optimal | Historical | Optimal | Historical | Optimal |





| | | | | | | |
|---|---|---|---|---|---|---|
| LYH user | 5.04 | 0.00 | 10.94 | 0.00 | 2.78 | 2.68 |
| JBHD user | 0.52 | 0.00 | 0.00 | 0.00 | 0.00 | 0.00 |
| SH user | 1.97 | 0.92 | 0.00 | 0.00 | 0.52 | 0.52 |
| GLJ-HA user | 3.98 | 1.78 | 11.28 | 1.91 | 2.69 | 3.40 |
| HZH user | 7.07 | 6.19 | 8.35 | 0.38 | 8.66 | 6.70 |
| FHH user | 17.22 | 2.59 | 14.41 | 0.00 | 16.64 | 1.95 |
| LYG user | 7.44 | 2.15 | 1.68 | 0.00 | 6.10 | 3.82 |
| EH-HY user | 3.11 | 3.09 | 1.73 | 0.50 | 6.06 | 4.55 |
| S-Z user | 11.28 | 5.67 | 12.46 | 3.13 | 9.41 | 7.64 |
| S-S user | 1.02 | 0.32 | 1.39 | 0.49 | 1.34 | 0.74 |
| S-P user | 3.62 | 1.35 | 4.15 | 1.55 | 5.11 | 3.08 |
| LMH user | 0.58 | 0.12 | 2.69 | 2.37 | 1.35 | 2.53 |
| WSH user | 2.91 | 3.21 | 6.10 | 3.16 | 2.55 | 3.64 |
| Variance | 22.87 | 4.16 | 26.18 | 1.51 | 21.54 | 5.01 |
| Decrement | 81.82% | | 94.24% | | 76.72% | |


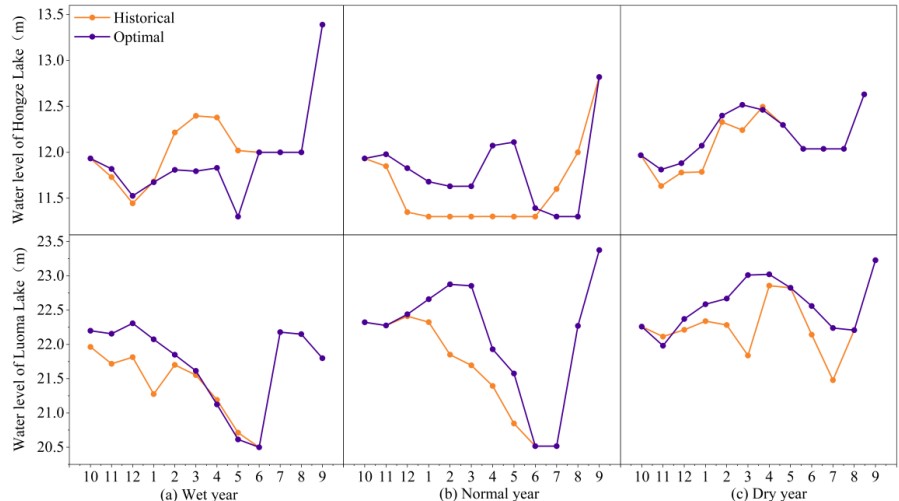


**Fig. 6. The operation water level of the Hongze and Luoma lake for (a) wet, (b) normal, and (c) dry years**
**under historical (orange line) and optimal (purple line) operation.**
Fig. 6 shows the water level variations of HZ Lake and LM Lake (two major storage lakes in
Jiangsu Province). The water level of HZ Lake and LM Lake is slightly higher than the historical value
after optimization for most of the time. Both lakes show a regular trend of change: the lake remains at a
high level during the dry season, releases water before the flood season, then remains at a low level
throughout the flood season for storing flood, and returns to a high level at the end of the flood season.
Therefore, the water transmission system can simultaneously implement lake water storage in the flood



and non-flood seasons to ensure the fulfilment of water supply tasks and improve the utilization
efficiency of water resources.

The optimized water level of HZ Lake was significantly lower than the historical value from

February to June in Fig. 6 (a). This is due to the large amount of natural precipitation with uneven
spatial-temporal distribution in the wet year, and the discharge time and discharge volume of HZ Lake
are unreasonable in historical operating strategy, resulting in a large amount of water deficit and
abandonment. Therefore, in the optimized operation strategy, a certain amount of lake water is
discharged during the impoundment period for water supply. Similarly, LM Lake level in April-June
below the historical level. In the Fig. 6 (b) and (c), the optimized HZ Lake level is significantly higher
than the historical value before the flood season, while the flood season is decreased dramatically,
indicating that the lake stores a certain amount of water while supplying water during the impoundment
period, and discharges water before the flood season to ensure that the flood season is at a safe level.
Similarly, for LM Lake, the water level increased after optimization in typical years, and tended to be
consistent after the flood season.

Compared with the historical operation strategy, the water level of HZ Lake and LM Lake is

slightly higher than the historical value after optimization for most of the time. Combined with the
results of Table 4 and Table 5, the coordinated water diversion between LM Lake and HZ Lake
increased obviously while reducing TWD and PW.
**Table 6 The water distribution of different routes between two adjacent lakes.**

| Inflow/ outflow Lake | Inflow/ outflow Route | Multi-year average | |
|---|---|---|---|
| | | Historical | Optimal |
| Outflow LM Lake | The ratio of inflow Hanzhuangyun and Bulao River | 19: 8 | 1: 1 |
| Inflow LM lake | The ratio of outflow Zhongyun and XuhongRiver | 7: 9 | 7: 1 |
| Outflow HZ Lake | The ratio of inflow Xuhong and Zhongyun River | 1: 6 | 1: 14 |
| Inflow HZ Lake | The ratio of outflow Liyun River and Jinbao channel | 19: 1 | 25: 1 |
| Water withdrawn from Yangtze River | The ratio of inflow Jinbao channel and Liyun River | 8: 9 | 2: 9 |

This paper considers that the water distribution of different routes between two adjacent lakes is



an important factor affecting the water deficit in the users. The water transmission capacity of the route is mainly affected by the water demand of users and the pumping capacity of the pumping station along the way. Table 6 shows the distribution of different water transmission routes between neighboring lakes in typical years. The ratio of water pumped from LM Lake into Hanzhuangyun River and Bulao River is decreased from 19: 8 to 1: 1, which is convenient for simultaneous double-route water supply outside the province and reduces the water supply pressure of Hanzhuangyun River. The ratio of water from the Zhongyun River and the Xuhong River to the LM Lake increased from 7: 9 to about 7: 1, gradually shifting the focus of the water transmission route to the Zhongyun River. The reason is that Hongze Lake transports water to the Zhongyun River through the Erhe and Gaoliangjian Sluice without pumping, which can reduce the pumped water and save project cost. In addition, there are many users in Zhongyun River, the water demand is higher. When the water supply is sufficient, the drainage through Huaiyin, Yanhe and Yangzhuang Sluice significantly reducing the water deficit of FHH and LYG users (see Table 5). Similarly, the ratio of water from HZ Lake to Zhongyun River and Xuhong River increased from 6:1 to 14:1. The ratio of pumping water from Liyun River and Jinbao Waterway to HZ Lake also increased from 19:1 to 25:1, indicating that the water transport efficiency of Zhongyun River is higher, and the operation strategy should be based on water transfer from the central canal, with the western route as the auxiliary support route. Similarly, the ratio of pumping water from the Yangtze River to the Liyun River and the Jinbao Channel has also increased (see Table 6).

Evidence from this research suggests that the water extracted from Hongze Lake is much greater than that from other lakes, indicating that Hongze Lake is the main source of water to support water supply and flood control within the system. Therefore, it is important to understand the water storage period. It is important to ensure that the water level of each storage lake reaches the water level at the end of the flood season to complete the water allocation in the non-flood season. If the reservoir water storage is insufficient, the Yangtze River will be pumped in time to ensure the normal operation of the entire water diversion project. The Yangtze River is the main source of water outside the J-SNWDP system. The proportion of pumping water from the Yangtze River in the pumping water volume of the system plays an important regulatory role. The actual operation should follow the following rules: When the natural inflow is less, mainly through pumping the Yangtze River to complete the task of water supply; when the natural inflow is large, the water withdrawn from Yangtze River is reduced, and the focus of water allocation is shifted to the mutual replenishment between lakes.





**4 Discussion**
The joint optimal operating model is operated based on social demand (water deficit) and
economic (pumped water) objectives, focusing on the issue of water deficit concentration. Herein, the
limited available water is used to minimize the total water deficit of the system and water deficit
differences between users and applies to inter-basin water transfer projects with complex systems and a
large number of water users. The multi-attribute decision implemented in this study incorporates
ecological (the abandoned water) and the water withdrawn from Yangtze River into the multi-attribute
decision indicator set, which can provide optimal operation strategy with preferred weights for decision
makers who have different preferences.
From the historical operation strategy, it can be seen that the water deficit doesn't only occur under
less natural inflow conditions, but also there are still serious problems of water deficit and
abandonment in wet years. The main reasons for the coexistence of water deficit and water
abandonment are lakes' limited water storage capacity and the uneven spatial-temporal distribution of
natural inflow. Water deficit and the pumped water are greatly affected by natural inflow; we should
not expect to find a general operating strategy optimal in all natural conditions. This paper performs
well in water resources allocation and utilization on a monthly time step of three typical years (wet year,
normal year and water shortage year), which is representative and universal, and provides a useful
guide for IBWD under future uncertainty. Therefore, in addition to implementing the optimal operation
strategy, the flood control limit water level should be appropriately increased according to the natural
inflow to improve the lake's storage capacity in the flood season. This is a potential way to effectively
protect the water diversion function of the project.
**5 Conclusions**
As the largest inter-basin water transfer project in the world, the South-to-North Water Transfer
Project, scientifically operating decisions are important for improving the water allocation balance and
reducing the stress of concentrated water deficits.
1) From the perspectives of social demand, economy and ecology, this paper establishes a joint optimal

operation model for the Jiangsu section of the South-to-North Water Diversion Project (J-

SNWDP), and further uses a combination of NSGA-III algorithm and multi-attribute decision-





making for strategy preference, which has a certain persuasion. This method has a good performance in solving the complex water transfer problems with multiple objectives and engineering units, and is currently less applied.

2) After incorporating the Water Deficit Evenness Index into the joint optimal operation model, the concentration of water deficit is reduced by 94.2% (81.8 %, 76.7%) in typical wet years (normal year and dry year) compared with the historical strategy, which greatly ameliorated the engineering problem of user water deficit concentration. The other two indicators of the model, total water deficit (TWD) and pumping water (PW), were reduced by 82.1% (37.7%, 52.4%) and 45.1% (3.2%, 21.5%), respectively, with excellent performance.

3) After optimization, the storage capacity of the lakes is enhanced, and the water allocation between different water transmission routes is more balanced, which improves water utilization and water supply efficiency. It puts forward the potential ways to effectively guarantee the water diversion task, providing the scientific basis and operating suggestions for the J-SNWDP.



**Acknowledgments**
This study was supported by the Jiangsu Province Water Science and Technology Key Projects

(2020005).

**Data availability**
Data will be made available on request.
**Author contribution**
Bingyi Zhou: Writing- Original Draft, Validation, Formal analysis, Software, Methodology,
Conceptualization, Visualization, Term Definition
Guohua Fang: Writing- Review and Editing, Funding acquisition, Resources, Supervision, Project
administration
Xin Li: Resources, Writing- Review and Editing, Visualization, Investigation.
Jian Zhou: Writing- Review and Editing, Supervision, Formal analysis.
Huayu Zhong: Resources, Data Curation, Visualization, Investigation.
**Competing interests**
The authors declare that they have no known competing financial interests or personal
relationships that could have appeared to influence the work reported in this paper.













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
