# Peer review of "Joint optimal operation of the South-to-North Water Diversion Project considering the evenness of water deficit"

_Hydrology and Earth System Sciences, 2023_

## Referee Comment (RC2)

Reviewer: The content is very relevant, as inter-basin water transfer (or transboundary systems) can alleviate the water deficit crisis caused by the uneven distribution of water resources. This new approach/index (i.e., Water Deficit Evenness Index – WDEI) seems to be a very efficient manner to sharing the pressure of water scarcity as a social demand objective, although some uncertainty in this methodology may be still relevant. Additionally, the English of the manuscript seems to be very good, congratulations. Here below and attached are some recommendations.

**Introduction**

Lines 32 – 34: Some references should be included in order to support your statement in these lines. I would recommend regional studies.

Lines 37 – 38: I would include more studies in this statement "Inter-basin Water transfer projects have been widely constructed worldwide (references)". I would recommend adding the references below, as they provide some examples of how the inter-basin water transfer impacted the water availability in these water demand regions. Or you could include them in Line 44, in order to support your statement.

Medeiros, P., & Sivapalan, M. (2020). From hard-path to soft-path solutions: slow–fast dynamics of human adaptation to droughts in a water scarce environment. Hydrological Sciences Journal, 65(11), 1803-1814.

Ghoreishi, M., Elshorbagy, A., Razavi, S., Blöschl, G., Sivapalan, M., & Abdelkader, A. (2022). Cooperation in a Transboundary River Basin: a Large Scale Socio-hydrological Model of the Eastern Nile. Hydrology and Earth System Sciences Discussions, 2022, 1-24.

Wei, Y., Wei, J., Li, G., Wu, S., Yu, D., Ghoreishi, M., ... & Tian, F. (2022). A socio-hydrological framework for understanding conflict and cooperation with respect to transboundary rivers. Hydrology and Earth System Sciences, 26(8), 2131-2146.

Lu, Y., Tian, F., Guo, L., Borzì, I., Patil, R., Wei, J., ... & Sivapalan, M. (2021). Socio-hydrologic modeling of the dynamics of cooperation in the transboundary Lancang–Mekong River. Hydrology and Earth System Sciences, 25(4), 1883-1903.

Lines 58 – 60: "Currently, most IBWD projects primarily follow various laws, regulations, policy guidelines, and historical experience in dispatching strategies set by the government." Which laws, regulations, or policy guidelines are those? Please, explain a little bit of them or

include a reference, only mentioning that there are a bunch of laws it is not so clear, you should show them in case the reader wants to know more about them in China.

Lines 60 – 62: What are the main lacks? You did mention that there a lack in the detailed operation rules, but you did not explain what this/these lack (s) is/are specifically. In addition, what is the main question (s) you want to address in this manuscript? These should be explicit in the introduction.

Lines 55 – 86; Lines 87 – 114; I would recommend breaking these long paragraphs in shorter ones, as sometimes hard to follow the main idea you wanted to transmit.

The introduction section provides a lot of information showing some of state-of-the-art studies (which is good), however, it needs to be shortened, and a deep reformulation is required. I would recommend maximum of four or five shorten paragraphs, with the last one being the objectives of this study.

Lines 126 – 127 are not necessary, however, I would recommend including a flowchart figure in the methodology section, showing which steps were required to implement this methodology, since the data gathering to the model/index approach execution.

**Methodology**

Figure 1: All the symbols in Figure 1 should be in the legend. It is a good idea to color the legends from each line (makes it easier to understand for the reader), but the symbols should be included as well, for example, the pumping and city symbols. In addition, what is the green line in the map? The coordinates from the left and top should be removed as they are just repeating information. In addition, I would make a zoom out of China, or maybe creating another box locating China in Asia continent, as although China's locations is well known, it is always good to include a broader map of the country we are studying.

Figure 3: I would suggest to change the color of the normal years to black, and the dry years to red, as it makes easier for the reader to follow. I would also recommend changing the rage of the Monthly mean inflow in Hongze Lake (a) to 0, 100, and 200, in Luoma Lake (b) to 0, 25,

and 50, in Nansi Lake (c) to 0, 1, and 2, respectively, that way your graphs would be cleaner. The Monthly mean average dots should be bigger as we can barely see them in the graphs.

Line 194: "The issue is particularly severe here." Here where?

Line 208: Where are the symbols meaning for Equation 1? As well as for the other used equations? Every equation should come with the meaning of each used symbol right after the equation.

Figure 4: I would recommend using the standard flowchart boxes/symbols. For example, rectangles means a "process", circle means a "terminal", and so on. That makes easier for the reader to understand what has been done in each step. This recommendation should also be applied to the recommended flowchart and for Figure 2.

**Results/Discussion**

Although your results showed to be very aligned with the necessities of one of China's water transboundary systems, was this methodology used elsewhere? If not, would it be possible to apply this methodology to other similar water transfer systems? Which would be the main concerns in applying this method in other locations? On top of that, what are the main uncertainties of this method? I mean, although your results showed a high confidence, I think there are some concerns in this methodology that should be considered and discussed.

Please add a new section regarding the "Sources of Uncertainty" as they were not so clear in the discussion section. The objective of this section is an auto critique of your own work, showing its limitations and it can be improved. No work is so perfect that can't show any limitations, please rethink about the main uncertainties and write this section.

In the discussion section, I would also recommend adding some statements showing the costs to implement and maintain the inter-basin water transfer, showing how much it is (in average) to install in China and comparing/discussing with other system worldwide (values in American dollar).

Another questions arise for the discussion section:

How many years did it last to be build? How many years it is expected to work? What are the main trade-offs (e.g., environment impacts; positives and negatives for the biota; how the local hydrology will be changed; how many energy is used to pump the water in the whole system; etc.) of this kind of system?

Are isolated communities allowed to use this transferred water where the transboundary system flows? If not, how this issue would be addressed?

Are there any conflicts between two communities (i.e., the one that is providing the water and the one that is receiving this resource)?

**Conclusion**

I suggest joining all points in the conclusion section. In my opinion it makes this section more "fluid" and easier to read. In addition, some recommendations for future work need to be included, for example, the application of this method in other similar watersheds worldwide in order to revalidate this method in other circumstances.

Overall, the work is very good, it just needs more discussion and some explanations, as I included in the review. The writing is also fine and formal.

---

## Author Comment (AC1)

**Ref: No. hess-2023-228**. "Joint optimal operation of the South-to-North Water Diversion Project considering the evenness of water deficit" by Zhou et al.

November 28, 2023

**Dear RC #1,**

We thank you and the anonymous reviewers for the constructive comments and suggestions on our manuscript entitled "Joint optimal operation of the South-to-North Water Diversion Project considering the evenness of water deficit" (**No. hess-2023-228**).

We have thoroughly revised the original manuscript accordingly. All of the changes we have made are marked in red in the revised manuscript. Point-by-point responses to the comments are attached to this letter. Our responses to comments are marked in **bold blue**. The original manuscript cited is indicated in *black italics*, and the modified text is shown in *red italics*. We hope that you will find this updated manuscript to your satisfaction and consider it for publication in *Hydrology and Earth System Sciences*.

Thank you for taking the time to consider our research and we look forward to hearing from you.

Sincerely,
Prof. Dr. Guohua Fang

College of Water Conservancy and Hydropower Engineering,
Hohai University, Nanjing 210098, P. R. China
E-mail: hhufgh@163.com

**Reply to reviewer' comment**

**Reviewer #1: Manuscript Review**

**Overall comment:**

This manuscript proposed an optimal operating strategy with a focus on alleviating the concentrated water shortage problem of the Jiangsu section of the South-to-North Water Diversion Project. As the authors stated, it is important to balance multiple objectives including social, economic, and ecological objectives, and find the optimal operation rules for efficient water resources utilization. The strategies derived from NSGA-III have been further filtrated with the multi-attribute decision-making method, which is quite innovative and can solve the problem of uneven spatial and temporal distribution of water resources. In general, this manuscript is organized well. I have several suggestions before it can be accepted for publication in Hydrology and Earth System Sciences.

**Response:**

**Thank you very much for your recognition of our work. You have provided us with very valuable comments to improve the quality of this paper. We have tried our best to digest your comments and made corresponding improvements carefully. Point-by-point responses are attached below.**

**Comment 1.** The Introduction section needs to be rewritten/reorganized to be more clear and focused. For a logically sound presentation, this section should start with an overall introduction of IBWD, followed by a generalized discussion of the main research efforts and remaining gaps of IBWD worldwide. Next, China's SNWDP-specific information should be provided in detail, which transitions to the key research objectives and contributions of this paper. With each of the above four aspects occupying at least one dedicated paragraph, I suggest proceeding straight to the point (SNWDP-focused study) after necessary background information (IBWD in general) to avoid blurring of the key message conveyed by the present study.

**Response 1:**

**Thank you for your suggestion. We have reorganized the Introduction section to make it clearer and more appropriately highlight the research focus of the paper. The updated Introduction are shown as follows:**

[revised manuscript text omitted]

**Comment 2**. Line 15, "NSGA" as an acronym should be defined on its first use in the manuscript.

**Response 2:**

**Thank you for your suggestion, we added the full name of "NSGA" in the manuscript upon its first appearance:**

*"Further, the Nondominated Sorting Genetic Algorithm III (NSGA-III) and multi-attribute decision-making were applied to solve the model and obtain an optimal operation strategy." (Lines 15-16 in updated manuscript; Lines 15 in old version)*

**Comment 3**. Lines 72-74, please add appropriate citations for the sentence "At the same time...".

**Response 3:**

**Thank you for your suggestion. We have added an appropriate citation for this sentence "At the same time..." to indicate the amount of abandoned water and the water withdrawn from the river as important indicators.**

*"At the same time, the proportion of the amount of abandoned water and the water withdrawn from the river in the process of water diversion should be taken as secondary considerations (Guo et al., 2018)." (Lines 59-61 in updated manuscript; Lines 72-74 in old version)*

**Comment 4**. Line 129, the title should be "2.1 Study area and data"

**Response 4:**

**Thanks for your comments. We have revised the title of Section 2.1. The revisions are shown as follows:**

*"2.1 Study area and data" (Lines 102 in updated manuscript; Lines 129 in old version)*

**Comment 5**. Line 192, the title should be "2.2 Water deficit evenness index"

**Response 5:**

**Thanks for your comments. We have revised the title of Section 2.2. The revisions are shown as follows:**

*"2.2 Water deficit evenness index" (Lines 165 in updated manuscript; Lines 192 in old version)*

**Comment 6**. Section 2.4 lacks the introduction of the specific application of the NSGA-Ⅲ algorithm in the model. In addition, why use the multi-attribute decision to filter again after obtaining a set of operation strategies? Please add the explanation.

**Response 6:**

**Thanks to your suggestion. We have added the introduction of the specific application of the NSGA-III algorithm, and additional information on the reasons for using multi-attribute decision making to filter again.**

*The NSGA-Ⅲ is used to solve the joint optimal operation model of the J-SNWDP under three typical conditions: wet year, normal year and dry year. The model takes the actual value of the pumped water of each pumping station as the decision variable. The population size, generation, crossover rate and mutation rate are set to 200, 20000, 0.9 and 0.1, respectively. The application of NSGA-III to the model can be summarized as the following steps:(1) Take 12 monthly water pumped of a year as the decision variables and initializes the population of size N based on the physical constraints of operation; (2) Calculate the objectives of TWD, EWD and WP for each chromosome and sort by Nondominated strategy; (3) Select excellent chromosome from population non-dominated as parent chromosome and create child chromosome by the cross and mutation operation. (4) Combine the parent chromosome with the child chromosome and update the population by Nondominated strategy. (5) Repeat the four steps above until the number of iterations is reached.*

*After solving the model using NSGA-III, we obtained a set of optimized running strategies (i.e., strategies based on the Pareto-ranked top 200). In order to measure the operation effect more comprehensively, more indicators are needed to assist in selecting an optimal operation strategy. Therefore, this paper further applies multi-attribute decision-making methods to screen and determine the optimal operation strategy. The abandoned water reflects the regulation and storage capacity of the lakes, and the water withdrawn from the Yangtze River reflects the impact of the water transferred outside the system on the operation strategy."* **(Lines 248-265 in updated manuscript; Lines 264-269 in old version)**

**Comment 7**. The text in Fig 4 is not clear, please revise the figure with a dark text color.

**Response 7:**

**Thank you for your suggestion. To avoid duplication and confusion, we have modified Figure 4 to include the original content of the complete optimization process flowchart. The revisions are shown as follows:**

"

[Figure]

*Fig. 4. Optimization process of the Jiangsu section of the South-to-North Water Diversion Project (J-SNWDP)." (Lines 272-274 in updated manuscript; Lines 277-279 in old version)*

**Comment 8**. The study is conducted to address the problem of concentrated water shortage. It is recommended that section 3.2 be specifically divided into several subsections based on the optimization effects from different aspects, so as to highlight the contribution of the Water deficit evenness index.

**Response 8:**

**Thank you for your comments. We have divided section 3.2 into 4 subsections to illustrate the comparison of the optimal operation strategy in different typical years with the historical operation strategy from 4 aspects, including the main operation performance indicators, the water deficit in water users, the operation water level of the Hongze and Luoma lake, and the water distribution of different routes between two adjacent lakes.**

*"3.2.1 Comparison of the main operation performance indicators*

*Table 4 shows the …… economic benefits.*

*3.2.2 Comparison of water deficit in water users*

*The comparison of …… general public.*

*3.2.3 Comparison of the operation water level of the Hongze and Luoma lake*

*Fig. 6 shows the …… TWD and PW.*

*3.2.4 Comparison of the water distribution of different routes between two adjacent lakes*

*This paper considers that …… the Jinbao Channel has also increased (see Table 6)." .*
*(Lines 325-416 in updated manuscript; Lines 330-430 in old version)*

**Comment 9**. Lines 419-430, the authors summarized the control rules for appropriate lake levels and proposed suggestions for actual operation, which is more suitable for the discussion section. Please revise and reorganize the discussion section.

**Response 9:**

**Thank you for your suggestion. We have removed paragraph " Evidence from…" and reorganized the discussion section to make it clear. Some parts of the revisions are as follows:**

*"4 Discussion*

[revised manuscript text omitted]

**Comment 10**. The use of (1) or 1) is not uniform throughout the manuscript. Please revise it.

**Response 10:**

**We have revised and unified all similar expressions in the manuscript, which can be found in the latest manuscript.**

**Comment 11**. Is this method universal? Are the results of this research capable of being applied to other similar cases? Please add some necessary sentences to illustrate this clearly in the conclusion.

**Response 11:**

**Thank you for your suggestion. To avoid any confusion, we have added a statement about the method's universality in the conclusion. The revisions are shown as follows:**

*"(3) After optimization, the rising trend of water level in Hongze Lake and Luoma Lake reflects the enhanced storage capacity of the lake, and the water allocation between different water transmission routes is more balanced, which improves water utilization and water supply efficiency. Moreover, this paper proposes water transfer prioritization rules and suggests appropriately increasing the flood control limit water level, aimed at protecting the water diversion and enhancing operational efficiency. The sources of uncertainty, such as natural inflow and societal water demand, are worthy of further study.*

*Overall, the successful application of the optimal operation strategy in the Jiangsu section of the South-to-North Water Transfer Project also demonstrates the feasibility of the research. It is hoped that this method can be attempted in other similar watersheds worldwide in order to revalidate this method in other circumstances, demonstrate its universality. This would provide the scientific basis and operating suggestions for the inter-basin water diversion project."* **(Lines 502-513 in updated manuscript; Lines 468-471 in old version)**

**Comment 12**. There are a few typos and grammar errors in the manuscript. English should be improved.

**Response 12:**

**Thanks to your suggestions, we have checked and revised the entire manuscript for typos and grammar.**

**Reference**

[revised manuscript text omitted]

---

## Author Comment (AC2)

**Ref: No. hess-2023-228**. "Joint optimal operation of the South-to-North Water Diversion Project considering the evenness of water deficit" by Zhou et al.

November 28, 2023

**Dear RC#2,**

We thank you for the constructive comments and suggestions on our manuscript entitled "Joint optimal operation of the South-to-North Water Diversion Project considering the evenness of water deficit" (**No. hess-2023-228**).

We have thoroughly revised the original manuscript accordingly. All of the changes we have made are marked in red in the revised manuscript. Point-by-point responses to the comments are attached to this letter. Our responses to comments are marked in **bold blue**. The original manuscript cited is indicated in *black italics*, and the modified text is shown in *red italics*. We hope that you will find this updated manuscript to your satisfaction and consider it for publication in *Hydrology and Earth System Sciences*.

Thank you for taking the time to consider our research and we look forward to hearing from you.

Sincerely,

Prof. Dr. Guohua Fang

College of Water Conservancy and Hydropower Engineering,

Hohai University, Nanjing 210098, P. R. China

E-mail: hhufgh@163.com

**Reply to reviewer' comment**

**Reviewer #2: Manuscript Review**

**Overall comment**

The content is very relevant, as inter-basin water transfer (or transboundary systems) can alleviate the water deficit crisis caused by the uneven distribution of water resources. This new approach/index (i.e., Water Deficit Evenness Index – WDEI) seems to be a very efficient manner to sharing the pressure of water scarcity as a social demand objective, although some uncertainty in this methodology may be still relevant. Additionally, the English of the manuscript seems to be very good, congratulations. Here below and attached are some recommendations.

**Response:**

**Thank you very much for your recognition of our work, you have provided us with very valuable comments to improve the quality of this paper. We have tried our best to digest your comments and made corresponding improvements carefully. Point-by-point responses are attached below.**

**Comment 1.** Lines 32 – 34: Some references should be included in order to support your statement in these lines. I would recommend regional studies.

**Response 1:**

**Thanks to your comments. We have reorganized the statement of the mentioned lines and provided some references for clarity. The revisions are shown as below:**

*"Influenced by the impacts of global climate change, human activities, and increasing water demand, issues like regional water resource deficits, flood and drought disasters, and the conflicts between water supply and demand are progressively intensifying (Florke et al., 2018; Kato and Endo, 2017; Ma et al., 2020; Rossi and Peres, 2023). These social issues have become one of the key factors constraining regional and even global sustainable development and environmental protection (Li et al., 2020; Liu et*

*al., 2021; Tian and Destech Publicat, 2017). Inter-basin Water transfer projects have been widely constructed worldwide as an effective way to address water deficit issues caused by uneven distribution of water resources, and improve their utilization efficiency (Medeiros and Sivapalan, 2020; Sun et al., 2021; Wei et al., 2022)."* **(Lines 31-38 in updated manuscript; Lines 32-39 in old version)**

**Comment 2.** Lines 37 – 38: I would include more studies in this statement "Inter-basin Water transfer projects have been widely constructed worldwide (references)". I would recommend adding the references below, as they provide some examples of how the inter-basin water transfer impacted the water availability in these water demand regions. Or you could include them in Line 44, in order to support your statement.

Medeiros, P., & Sivapalan, M. (2020). From hard-path to soft-path solutions: slow–fast dynamics of human adaptation to droughts in a water scarce environment. Hydrological Sciences Journal, 65(11), 1803-1814.

Ghoreishi, M., Elshorbagy, A., Razavi, S., Blöschl, G., Sivapalan, M., & Abdelkader, A. (2022). Cooperation in a Transboundary River Basin: a Large Scale Socio-hydrological Model of the Eastern Nile. Hydrology and Earth System Sciences Discussions, 2022, 1-24.

Wei, Y., Wei, J., Li, G., Wu, S., Yu, D., Ghoreishi, M., ... & Tian, F. (2022). A sociohydrological framework for understanding conflict and cooperation with respect to transboundary rivers. Hydrology and Earth System Sciences, 26(8), 2131-2146.

Lu, Y., Tian, F., Guo, L., Borzì, I., Patil, R., Wei, J., ... & Sivapalan, M. (2021). Sociohydrologic modeling of the dynamics of cooperation in the transboundary Lancang–Mekong River. Hydrology and Earth System Sciences, 25(4), 1883-1903.

**Response 2:**
**Thank you for your comments. We have included and clarified the references you**

**listed in the appropriate places to support our statements about the implementation of inter-basin water transfer projects worldwide. The revisions are shown as below:**

*"Inter-basin Water transfer projects have been widely constructed worldwide as an effective way to address water deficit issues caused by uneven distribution of water resources, and improve their utilization efficiency (Medeiros and Sivapalan, 2020; Sun et al., 2021; Wei et al., 2022). At least 10 % of the cities worldwide receive water from IBWD projects (McDonald et al., 2014). The birth of the Lancang-Mekong Cooperation promotes the joint development of six countries, namely China, Cambodia, Laos, Myanmar, Thailand and Vietnam (Ghoreishi et al., 2023). The California State Water Project, the Colorado River Aqueduct (Lopez, 2018), the Senqu-Vaal transfer in South Africa and Lesotho (Gupta and van der Zaag, 2008), the Snowy Mountains Scheme in southeastern Australia (Pigram, 2000), and other inter-basin water transfer projects have all effectively alleviated water scarcity issues in various regions (Lu et al., 2021)."*
***(Lines 36-45 in updated manuscript; Lines 37-44 in old version)***

**Comment 3.** Lines 60 – 62: What are the main lacks? You did mention that there a lack in the detailed operation rules, but you did not explain what this/these lack (s) is/are specifically. In addition, what is the main question (s) you want to address in this manuscript? These should be explicit in the introduction.

**Response 3:**

**Thank you for your suggestion. In the updated manuscript, we have detailed the shortcomings of the current operational strategies and segregated this content into a standalone paragraph for a clearer presentation of the main issues that the manuscript aims to address. The revisions are shown as below:**

*"As the project continues to operate, the focus of research should be concentrated on the planning of operational strategies to enhance the sustainability of the project. For IBWT projects, due to regional differences, improving operational efficiency and*

*benefits while ensuring water supply is a challenging task. Currently, most IBWT projects primarily adhere to various laws and policies established by the government. These projects comply with annual water demand plans submitted by sectors like agriculture, domestic use, and ecology. The water supply principle is based on 'prioritizing users that are closer in distance, have lower water supply costs, and have larger water demands' to develop operation strategies. Such method of water diversion results in lower satisfaction levels for users that are farther in distance and have higher costs, leading to an imbalance in water supply and causing some users to face significant pressure from concentrated water deficits. Furthermore, these projects lack annual predictive assessments of local hydrological conditions and fail to develop targeted operational strategies for diverse natural inflows or extreme events. Developing operational strategies without considering the evenness of water deficit and natural inflows is unscientific. This inspires the primary objective of optimization in this paper."* **(Lines 65-78 in updated manuscript; Lines 60-62 in old version)**

**Comment 4.** Lines 55 – 86; Lines 87 – 114; I would recommend breaking these long paragraphs in shorter ones, as sometimes hard to follow the main idea you wanted to transmit.

**Response 4:**

**Thank you for your suggestion. We have reorganized the order and content of the mentioned for a clearer logic.**

*"There are considerable studies on the water resources operating strategy of the supply-oriented IBWT projects in terms of social, economic, ecological, and environmental (Gan et al., 2011; Liu and Zheng, 2002; Xu et al., 2013; Zhu et al., 2014). In general, meeting the water demand of various users is the main task of the IBWT project, with the consideration of minimizing water deficit in previous studies (Guo et al., 2020; Wang et al., 2008). Rather than the total amount of water deficit, the crux of the problem may actually be the concentration of water deficit in a certain period of time or region, which has not yet received sufficient attention and remains a major*

challenge. Therefore, both the total and spatial-temporal distribution of water deficit should be considered in the optimization process (Xu et al., 2013). In addition, users' demands and decision makers' benefits should be considered as priorities (Zhang et al., 2012), so minimizing pumped water (PW) is a direct way to reduce costs. *At the same time, the proportion of the amount of abandoned water and the water withdrawn from the river in the process of water diversion should be taken as secondary considerations (Guo et al., 2018).* However, due to the data on natural water and user water demand as the determining factors of the operation strategy, and the obvious regional differences, most of the objectives determined by the existing studies can only solve small-scale projects.

*As the project continues to operate, the focus of research should be concentrated on the planning of operational strategies to enhance the sustainability of the project. For IBWT projects, due to regional differences, improving operational efficiency and benefits while ensuring water supply is a challenging task. Currently, most IBWT projects primarily adhere to various laws and policies established by the government. These projects comply with annual water demand plans submitted by sectors like agriculture, domestic use, and ecology. The water supply principle is based on 'prioritizing users that are closer in distance, have lower water supply costs, and have larger water demands' to develop operation strategies. Such method of water diversion results in lower satisfaction levels for users that are farther in distance and have higher costs, leading to an imbalance in water supply and causing some users to face significant pressure from concentrated water deficits. Furthermore, these projects lack annual predictive assessments of local hydrological conditions and fail to develop targeted operational strategies for diverse natural inflows or extreme events. Developing operational strategies without considering the evenness of water deficit and natural inflows is unscientific. This inspires the primary objective of optimization in this paper.*

*The South-to-North Water Diversion Project (SNWDP) presents a highly complex and dynamic water situation, especially in the Jiangsu section (Vogel et al., 2015). Due to differences in the location and timing of natural inflows and water users, and the*

*aforementioned issues in operational strategies, an imbalance in water supply has arisen. At present, there have been some studies attempting to address this issue, but they tend to focus on meeting the total water demand and improving the overall benefits (Li et al., 2017; Zhuan et al., 2016), neglecting the fairness of water supply among different regions. Water supply may become concentrated on a specific user or time period. Therefore, it is of great theoretical significance and practical application value to optimize the existing operation strategy to alleviate the concentration of water deficit so as to realize the comprehensive benefits of the IBWT project (Nazemi and Wheater, 2015; Peng et al., 2015)."* **(Lines 50-88 in updated manuscript; Lines 55-114 in old version)**

**Comment 5.** The introduction section provides a lot of information showing some of state-of-the-art studies (which is good), however, it needs to be shortened, and a deep reformulation is required. I would recommend maximum of four or five shorten paragraphs, with the last one being the objectives of this study.

**Response 5:**

**Thank you for your suggestion. We have shortened and restructured the introduction to make the research content of the article more concisely and clearly. The updated Introduction section is shown as follows:**

[revised manuscript text omitted]

**Comment 6.** Lines 126 – 127 are not necessary, however, I would recommend including a flowchart figure in the methodology section, showing which steps were required to implement this methodology, since the data gathering to the model/index approach execution.

**Response 6:**

**Thank you for your suggestion. We have removed lines 126-127 and replaced Figure 4 in the "Methods" section with a flowchart showing the complete optimization process. The revisions are shown as follows:**

"

[Figure]

*Fig. 4. Optimization process of the Jiangsu section of the South-to-North Water Diversion Project (J-SNWDP)." (Lines 272-274 in updated manuscript; Lines 277-279 in old version)*

**Methodology**

**Comment 7.** Figure 1: All the symbols in Figure 1 should be in the legend. It is a good idea to color the legends from each line (makes it easier to understand for the reader), but the symbols should be included as well, for example, the pumping and city symbols. In addition, what is the green line in the map? The coordinates from the left and top should be removed as they are just repeating information. In addition, I would make a zoom out of China, or maybe creating another box locating China in Asia continent, as although China's locations is well known, it is always good to include a broader map of the country we are studying.

**Response 7:**

**Thank you for your suggestion. We have removed the duplicate coordinates in the Fig. 1 and added the symbols legend. In addition, the inset in the upper right corner has been changed to a geographic divisional map of Asia to show China's relative position in Asia. The green line in the figure indicates the West Route of the two water transmission routes. To avoid any confusion, we have added the legend of the green, orange and purple lines in the figure.**

"

[Figure]

*Fig. 1. The Jiangsu section of the South-to-North Water Diversion Project (J-SNWDP). **The orange, green and purple lines represent the Canal Route, the West Route, and intersection of the two routes to transport water outside the province, respectively.** " (Lines 143-146 in updated manuscript; Lines 170-173 in old version)*

**Comment 8.** Figure 3: I would suggest to change the color of the normal years to black, and the dry years to red, as it makes easier for the reader to follow. I would also recommend changing the rage of the Monthly mean inflow in Hongze Lake (a) to 0, 100, and 200, in Luoma Lake (b) to 0, 25, and 50, in Nansi Lake (c) to 0, 1, and 2, respectively, that way your graphs would be cleaner. The Monthly mean average dots should be bigger as we can barely see them in the graphs.

**Response 8:**

Thank you for your suggestion. We have revised and updated the range of the Monthly mean inflow, the color of the lines, and the size of the Monthly mean average dots. The revisions are shown as follows:

[Figure]

*Fig. 3. Monthly mean inflow curves of the (a) Hongze, (b) Luoma, and (c) Nansi lakes in typical wet, normal, and dry years." (Lines 160-164 in updated manuscript; Lines 187-191 in old version)*

**Comment 9.** Line 194: "The issue is particularly severe here." Here where?

**Response 9:**

**The problem here refers to the complex situation of multiple users, where the concentration of water deficits due to uneven water supply is particularly problematic. To avoid confusion, we have modified the statement in section 2.2.**

"For IBWT projects, issues of equitable water supply often arise, especially for projects like the J-SNWDP, which has 13 water users along its route. *Different users have significant spatial variations in location, as well as large differences in water supply costs. Consequently, the concentration of water deficits is particularly severe when there are multiple users, which results in some of them being stressed by water deficits.*"
*(Lines 166-170 in updated manuscript; Lines 193-194 in old version)*

**Comment 10.** Line 208: Where are the symbols meaning for Equation 1? As well as for the other used equations? Every equation should come with the meaning of each used symbol right after the equation.

**Response 10:**

**Thank you for your suggestion. We have added the symbols meanings right after all equations to avoid any confusion. The revisions are shown as follows:**

[revised manuscript text omitted]

**Comment 11.** Figure 4: I would recommend using the standard flowchart boxes/symbols. For example, rectangles means a "process", circle means a "terminal", and so on. That makes easier for the reader to understand what has been done in each step. This recommendation should also be applied to the recommended flowchart and for Figure 2.

**Response 11:**

**Thank you for your suggestion. To avoid duplication and confusion, we have modified Figure 4 to include the original content of the complete optimization process flowchart, which can be found in the Response 6.**

**Similarly, we have modified Figure 2. The reason why the lakes are all represented as circles with no absolute "terminal" in Figure 2 is that there is an exchange of water between the lakes.**

"

[Figure]

Fig. 2. Schematic diagram of the Jiangsu section of the South-to-North Water Diversion Project." (Lines 147-148 in updated manuscript; Lines 174-175 in old

*version)*

**Results/Discussion**

**Comment 12.** Although your results showed to be very aligned with the necessities of one of China's water transboundary systems, was this methodology used elsewhere? If not, would it be possible to apply this methodology to other similar water transfer systems? Which would be the main concerns in applying this method in other locations? On top of that, what are the main uncertainties of this method? I mean, although your results showed a high confidence, I think there are some concerns in this methodology that should be considered and discussed.

**Response 12:**

Thank you for your suggestions. Regarding the optimization objectives of this paper, the introduction highlights that the total water deficit and the pumping water are commonly used as optimization objectives in domestic and international discussion on inter-basin water transfer projects. However, the evenness of water deficit in this paper is rarely studied. Methodologically, the NSGA-III algorithm used in this paper is commonly employed as a solution method for solving models, and multi-attribute decision-making methods are often applied in indicator evaluation. Using this combination results is an innovative way to select the optimal operation strategies in a more scientific way.

The distribution of water resources varies significantly worldwide, and the concentration of water deficit is a prevalent and pressing issue that needs to be addressed in water-deficient areas. In addition to focusing on reducing the total water deficit, it is important to balance the allocation among water users to prevent the concentration of water deficit pressure in certain users. Therefore, we believe this method could be applied to other similar water transfer systems. When adapting it to different regions, a through exploration of local natural inflow patterns, topography, societal water demands, and economic development is essential to develop tailored optimal operation strategies.

For the main sources of uncertainty in the method, we have provided a detailed explanation in the "Sources of Uncertainty" section of this paper (see Response

**13), which highlights the concerns in this methodology that should be considered and discussed, as well as future research directions.**

**Comment 13.** Please add a new section regarding the "Sources of Uncertainty" as they were not so clear in the discussion section. The objective of this section is an auto critique of your own work, showing its limitations and it can be improved. No work is so perfect that can't show any limitations, please rethink about the main uncertainties and write this section.

**Response 13:**

**Thanks to your suggestion. We added a "Sources of Uncertainty" section to the limitations and areas for improvement of our study. The revisions are shown as follows:**

*"5 Sources of Uncertainty*

*This paper proposes an optimal operation strategy that considers the evenness of water deficit, accounting for the hydrological conditions of three typical years (i.e., wet years, normal years, and dry years). The strategy can generally alleviate the concentration of water deficit under most natural inflow conditions. However, due to the impact of global climate change, future runoff is highly uncertain, necessitating further discussion. Moreover, the Eastern Route of the South-to-North Water Diversion Project is a large-scale inter-basin water transfer project that spans provinces such as Jiangsu and Shandong, and it involves numerous uncertain factors. These include changes in biological communities, hydrological variations, and dynamic changes in water demand caused by extreme events, all of which add complexity to determining the water supply capacity of the project.*

*Regarding the two major regulating reservoirs within Jiangsu, Hongze and Luoma Lakes, the operational control water levels used in this paper were approved and established by the Ministry of Water Resources of China in 1954. Over the years, with socio-economic development, changes in the South-to-North Water Diversion Project, and the flood control capabilities of the lakes themselves, the original flood limit water*

*levels have become inadequate for current development needs. This paper proposes the idea of appropriately raising the flood limit water levels of the lakes during the flood season, but the specific values require further research, incorporating runoff forecasting, flood risk early warning, and other factors into future studies of inter-basin water transfer.*

*Furthermore, this paper places greater emphasis on the positive impacts of inter-basin water transfer on social demands and ecology, with less focus on the analysis of economic costs. Due to the variability in water prices across different regions and the ongoing changes in economic development in recent years, the cost of the project is currently represented by the volume of water pumped and has not been converted into actual cost prices. Future studies will delve into the dynamic adjustment mechanisms of water pricing, subsequently analyzing the economic benefits of the project."* **(Lines 460-483 in updated manuscript)**

**Comment 14.** In the discussion section, I would also recommend adding some statements showing the costs to implement and maintain the inter-basin water transfer, showing how much it is (in average) to install in China and comparing/discussing with other system worldwide (values in American dollar).

**Response 14:**

**Thank you for your suggestion. We have added a comparison of the costs of various inter-basin water transfer projects at the beginning of the discussion section to highlight the complexity and representativeness of the study area.**

*"Inter-basin water transfer projects are widely used around the world and are also quite costly to construct. For instance, the Colorado River aqueduct cost approximately 3.5 billion dollars (Witcher, 2017), the Australian Snowy Mountains Scheme was completed in 1974 at a cost of about 500 million dollars (Pigram, 2000), and the South-to-North Water Diversion Project in China, as the largest and most expensive inter-basin water transfer system in the world, is projected to cost 62 billion dollars (Markosov, 2014). The installation and operation maintenance costs are also*

*significantly high. Of this, the investment for just the Eastern Route of the project is around 1 billion dollars (Liu et al., 2022), which is a typical example of a vast and complex water transfer system and has certain representativeness. Therefore, the Jiangsu section of the Eastern Route of the South-to-North Water Diversion Project is selected as the study area for this paper."* **(Lines 418-427 in updated manuscript)**

**Comment 15.** Another questions arise for the discussion section:

How many years did it last to be build? How many years it is expected to work? What are the main trade-offs (e.g., environment impacts; positives and negatives for the biota; how the local hydrology will be changed; how many energy is used to pump the water in the whole system; etc.) of this kind of system?

**Response 15:**

**Thanks to your suggestion. The Eastern Route of the South-to-North Water Diversion Project (SNWDP) commenced construction in 2002 (Zhou et al., 2023). It officially began operation in 2013 (Wu et al., 2018) and has been in operation for 10 years to date, with a designed service life of 100 years. Since the 1950s, considering the severe water deficit in the northern regions and the urgent social needs, coupled with the Yangtze River in the southern region owning abundant water resources, the Chinese government invested significant efforts in planning the SNWDP, which is also of great importance in the project's construction.**

**The transfer of water resources also alters hydrological conditions and the characteristics of local water bodies, thus affecting the environment and ecology accordingly. In the early stages of construction, experts conducted rational and scientific water withdrawal assessments to ensure the minimization of negative environmental impacts due to the construction and operation of the project. As the project gradually enters the stable operational phase, it has been observed that water diversion has a significant positive effect on improving water quality and reducing sedimentation in the Yangtze River. (Yang et al., 2001; Zhang et al., 2022) Regarding the issue of energy consumption, the Middle Route of the SNWDP**

diverts water from the Danjiangkou Reservoir to Beijing and Tianjin through a newly excavated canal using gravity-driven water method, which eliminate the need for pumping and resulting in relatively low energy consumption. The East Route pumps water from Jiangsu to Shandong through 13 levels of pumping stations, and then it is gravity-driven from Shandong to Tianjin. The energy consumption generated by water pumping can be minimized through optimized scheduling (Liu et al., 2023). The reduction in pumping volume in the research results indirectly indicates an effective decrease in water pumping and relative energy costs.

**Comment 16.** Are isolated communities allowed to use this transferred water where the transboundary system flows? If not, how this issue would be addressed?

**Response 16:**

Thank you for presenting an interesting question. Recently, we have consulted with relevant departments of the Eastern Route of the SNWDP and investigated relevant information to conduct research on the issue.

Firstly, the division of the water users within Jiangsu Province by this project almost covers all cities, towns and villages. The allocation of water includes multiple water-using sectors such as agriculture, industry, domestic, navigation, and ecology. The annual water demand plans for each user and the water supply plans for the project are jointly formulated by the Ministry of Water Resources of China and the department of Water Resources of Jiangsu Province, which essentially ensure the water demands of all sectors of society, thus there are seldom isolated communities.

Furthermore, regarding water abstraction and usage issues, the Jiangsu section of the SNWDP is under regulation by the Jiangsu Provincial Government and the Water Resources Department. To ensure the rational use of water resources, the abstraction of water must follow a strict approval process. All the pumping stations along the route are monitored by dedicated personnel, and regular inspections are arranged. Even if there are isolated communities, unauthorized

**abstraction of water is not permitted in legal.**

**Comment 17.** Are there any conflicts between two communities (i.e., the one that is providing the water and the one that is receiving this resource)?

**Response 17:**

**The primary water sources (i.e., the providing communities) for inter-basin water transfer projects are mostly rivers or lakes with abundant water supplies. The process of diverting water from the providing communities to the receiving communities inevitably impacts the existing social and ecological balance. Considering the large scale of inter-basin water transfer projects and their impact on diverse natural and social conditions, the timing and volume of transfers undergo strict government deliberation to balance the water demands of receiving communities with minimal impact on all involved areas. Moreover, we have considered the water deficit evenness and inter-basin water transfers based on the principle of sharing the risk of water deficit in this paper. This approach also aims to prevent severe consequences caused by significant water deficits in a small area and to minimize potential conflicts between the supply and receiving communities.**

**Conclusion**

**Comment 18.** I suggest joining all points in the conclusion section. In my opinion it makes this section more "fluid" and easier to read. In addition, some recommendations for future work need to be included, for example, the application of this method in other similar watersheds worldwide in order to revalidate this method in other circumstances. Overall, the work is very good, it just needs more discussion and some explanations, as I included in the review. The writing is also fine and formal.

**Response 18:**

**Thank you for your suggestion. We have revised the conclusion and added content regarding the broader application of the method in similar studies and future works. The updated Conclusions are shown as follows:**

*"5 Conclusions*

[revised manuscript text omitted]

Liu, D.C., Li, Y., Wang, P.F., Zhong, H.Q., Wang, P., 2021. Sustainable Agriculture Development in Northwest China Under the Impacts of Global Climate Change. Frontiers in Nutrition 8. http://dx.doi.org/10.3389/fnut.2021.706552

Liu, Y., Chong, F.T., Jia, J.J., Cao, S.L., Wang, J., 2022. Proper Pricing Approach to the Water Supply Cost Sharing: A Case Study of the Eastern Route of the South to North Water Diversion Project in China. Water 14. http://dx.doi.org/10.3390/w14182842

Liu, Y.Y., Zheng, H., Wan, W.H., Zhao, J.S., 2023. Optimal operation toward energy efficiency of the long-distance water transfer project. Journal of Hydrology 618. http://dx.doi.org/10.1016/j.jhydrol.2023.129152

[revised manuscript text omitted]

Witcher, T.R., 2017. The Colorado River Aqueduct. Civil Engineering 87, 46-49. http://dx.doi.org/10.1061/ciegag.0001187

Wu, Y., Dai, R., Xu, Y.F., Han, J.G., Li, P.P., 2018. Statistical Assessment of Water Quality Issues in Hongze Lake, China, Related to the Operation of a Water Diversion Project. Sustainability 10. http://dx.doi.org/10.3390/su10061885

Xu, J.P., Tu, Y., Zeng, Z.Q., 2013. Bilevel Optimization of Regional Water Resources Allocation Problem under Fuzzy Random Environment. Journal of Water Resources Planning and Management 139, 246-264. http://dx.doi.org/10.1061/(asce)wr.1943-5452.0000248

Yang, S.L., Ding, P.X., Chen, S.L., 2001. Changes in progradation rate of the tidal flats at the mouth of the Changjiang (Yangtze) River, China. Geomorphology 38, 167-180. http://dx.doi.org/10.1016/s0169-555x(00)00079-9

Zhang, C., Wang, G.L., Peng, Y., Tang, G.L., Liang, G.H., 2012. A Negotiation-Based Multi-Objective, Multi-Party Decision-Making Model for Inter-Basin Water Transfer Scheme Optimization. Water Resources Management 26, 4029-4038. http://dx.doi.org/10.1007/s11269-012-0127-9

Zhang, T., Yang, G., Zhang, J., Wang, P., Chen, Y., Zeng, F., 2022. Changes in the Quality of Water Flowing Through the First Phase of the Eastern Route of the South-to-North Water Transfer Project. Journal of Hydroecology 43, 8-15.

Zhou, Y.Q., Chen, L.L., Zhou, L., Zhang, Y.L., Peng, K., Gong, Z.J., Jang, K.S., Spencer,

R.G.M., Jeppesen, E., Brookes, J.D., Kothawala, D.N., Wu, F.C., 2023. Key factors driving dissolved organic matter composition and bioavailability in lakes situated along the Eastern Route of the South-to-North Water Diversion Project, China. Water Research 233. http://dx.doi.org/10.1016/j.watres.2023.119782

Zhu, X.P., Zhang, C., Yin, J.X., Zhou, H.C., Jiang, Y.Z., 2014. Optimization of Water Diversion Based on Reservoir Operating Rules: Analysis of the Biliu River Reservoir, China. Journal of Hydrologic Engineering 19, 411-421. http://dx.doi.org/10.1061/(asce)he.1943-5584.0000805

Zhuan, X.T., Li, W., Yang, F., 2016. Optimal operation scheduling of a pumping station in east route of South-to-North water diversion project. ICAE 105, 3031-3037. http://dx.doi.org/10.1016/j.egypro.2017.03.623